# Beyond Invariance: A Feature-Strength Perspective for OOD Generalization

## Abstract

Out-of-Distribution (OOD) generalization is a central challenge in machine learning. Models often fail on unseen data, not because of an inability to learn robust signals, but because they *preferentially learn spurious, dataset-specific correlations that are highly predictive for in-distribution examples*. Existing solutions typically focus on searching for invariant features, yet often overlook a more fundamental question: **what properties of the training data cause models to learn these non-invariant "shortcut" features in the first place?** In this work, we present a different perspective on OOD generalization. We argue that failures to generalize are a direct consequence of models learning the strongest features in the training data, which are often spurious. Guided by this, we reframe OOD generalization not as a search for invariance, but as the *problem of identifying and mitigating the influence of these overly dominant features*. Under this new perspective, we develop a novel primitive for quantifying feature strength across a training set. This primitive gives rise to a targeted regularization algorithm that weakens a model's reliance on the identified strongest features, thereby compelling it to learn more robust and causally stable signals. Our method demonstrates substantial improvements in generalization across a wide range of OOD benchmarks, improving OOD accuracy by up to $2\times$ over standard training and significantly outperforming existing baselines without compromising in-distribution performance.

## 1 Introduction

A central goal of machine learning is to build models that generalize beyond their training data. Yet, the standard paradigm of minimizing empirical risk often fails precisely at this task, especially when faced with Out-of-Distribution (OOD) data (Beery et al., 2018; Ben-David et al., 2010; Bengio et al., 2019; DeGrave et al., 2020; Moreno-Torres et al., 2012; Recht et al., 2019; Taori et al., 2020). Models that achieve high in-distribution accuracy frequently do so by exploiting spurious correlations, *i.e.*, "shortcut" features (Geirhos et al., 2020; Pezeshki et al., 2021) that are highly predictive within the training set but fail to hold in new environments. A model trained to classify cows, for instance, might learn to recognize pastures rather than the animal itself; a medical diagnostic tool might rely on hospital-specific markings instead of the underlying pathology. These failures are not exceptions, but a common consequence of training powerful models on finite, biased datasets.

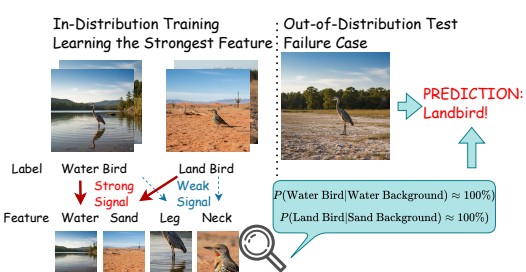

Figure 1: Illustration of spurious feature reliance: models trained with ERM often latch onto the statistically strongest signals (e.g., background water or sand) rather than robust causal features, leading to misclassification under distribution shift.

The prevailing perspective on mitigating such failures is to frame OOD generalization as a search for invariance. This approach, rooted in principles of causality, seeks to identify and isolate features whose predictive relationship with the label remains stable across different training environments. This perspective is a natural one and has led to a host of methods, including invariant risk minimization (Arjovsky et al., 2019; Ahuja et al., 2020; Koyama & Yamaguchi, 2020; Krueger et al.,

2021; Robey et al., 2021; Zhang et al., 2021a) and distributional matching (Ganin et al., 2016; Li et al., 2018; Sun & Saenko, 2016). The core assumption is that if a model can be restricted to only use these stable, invariant features, it will naturally generalize to unseen domains where the spurious correlations have shifted. However, while the pursuit of invariance is well-motivated, it often sidesteps a more fundamental question:

> **(Q)** *Why do models learn spurious, non-invariant features so readily in the first place, even when robust alternatives exist in the data?*

In this work, we take a step back and offer a different perspective on OOD generalization. We argue that the failures of modern classifiers are not merely an incidental byproduct of training, but a direct consequence of the learning objective itself, such as Empirical Risk Minimization (ERM) forces the model to learn the most statistically powerful signals to minimize the training loss, even if those signals are spurious "shortcuts" that fail to generalize. Models are designed to find the most predictive signals available (Ahuja et al., 2020; Heinze-Deml et al., 2018; Ahuja et al., 2021), and in many real-world datasets, the most statistically powerful features are precisely the spurious ones (see Figure 1). A simple background texture or a watermark, if consistently correlated with a label, provides a far easier-to-learn signal than the complex, nuanced features that define a class in a truly robust way. The optimization process, driven by empirical risk minimization, has no inherent mechanism to distinguish between a causal feature and a coincidental one; it only follows the path of steepest descent on the training loss.

The problem, then, is not simply that non-invariant features exist, but that they are often the strongest features in the training data; these are easy-to-learn, highly predictive signals that the model preferentially learns to minimize its training error. This reveals a fundamental tension: the very process designed to achieve high accuracy on the training set may be the primary driver of generalization failure on OOD data. This observation suggests a reframing of the OOD problem: rather than searching for invariant features, we should instead focus on identifying the features that are overly dominant and mitigating their influence. If a model's reliance on these strong-but-spurious signals can be weakened, it can be compelled to learn the more subtle, but more robust, features that are essential for true generalization.

Guided by this new perspective, we develop a simple yet effective framework for improving OOD generalization by directly addressing the influence of dominant features. We first introduce a novel primitive for quantifying the "strength" of any feature across the training set. This primitive measures the degree to which the presence of a feature in the training data influences the model's final predictions. This naturally gives rise to a detection and regularization algorithm. Our method first identifies the features with the highest strength scores and, by extension, the training examples that provide the most support for them. It then applies a targeted regularization that weakens the model's dependence on these specific examples. This process discourages the model from overfitting to the strongest, most obvious correlations and encourages it to learn the causally stable signals that generalize to new distributions. **In summary**, our contributions are as follows:

❶ We re-frame the problem of OOD generalization as one of detecting and mitigating the influence of the strongest features in a dataset, which are often spurious. We argue that generalization failures are a direct consequence of the learning objective itself, which compels models to preferentially learn these dominant, and often spurious, signals over more robust alternatives.

❷ Based on this new perspective, we introduce a novel primitive for quantifying the "strength" of any feature across a training set. This primitive enables an algorithm that first identifies the most dominant features and the training examples that support them, and then applies a targeted regularization to provably weaken the model's reliance on these specific examples, compelling it to learn more causally stable signals.

❸ We demonstrate the effectiveness of our approach through extensive experiments on OOD Benchmarks like *Waterbirds*, *Colored MNIST* (Lecun et al., 1998), *CelebA* (Liu et al., 2015), *Digits*, *PACs* (Li et al., 2017), and *VLCS* (Torralba & Efros, 2011). Our method achieves substantial gains, increasing OOD accuracy up to $2\times$ over standard ERM (Vapnik, 1991) and significantly outperforming strong baselines including IRM (Arjovsky et al., 2019), IB-ERM (Ahuja et al., 2021), Group-DRO (Sagawa et al., 2020), and CORAL (Sun & Saenko, 2016).

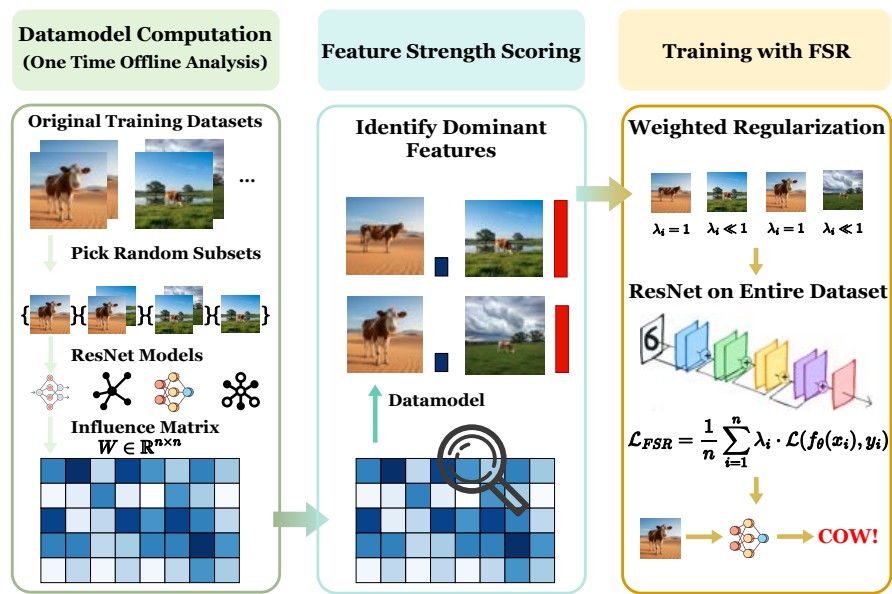

Figure 2: Overview of the proposed Feature Strength Regularization (FSR) framework. The method computes datamodels on random subsets to identify dominant features and applies weighted regularization to reduce reliance on spurious signals, guiding the model toward more robust representations.

## 2 RELATED WORKS

The problem of OOD generalization has motivated a significant body of research, largely centered on learning models that are robust to shifts between training and test distributions (Liu et al., 2021). A prevailing perspective casts this as a problem of invariance, seeking to isolate features that are causally linked to the label and thus stable across domains. Our work diverges from this tradition by proposing that the root cause of poor generalization lies not just in the existence of spurious features, but in their statistical strength within the training data. One prominent line of work, exemplified by Invariant Risk Minimization (IRM) (Arjovsky et al., 2019), attempts to learn representations that elicit an optimal classifier across multiple training environments. This principle is also explored in various other forms (Ahuja et al., 2020; Krueger et al., 2021; Jiang & Teney, 2024). While IRM and its variants seek to isolate features that are stable across domains, our approach addresses a more fundamental question: why models fail to learn them in the first place. We posit that models preferentially learn the strongest features, regardless of their invariance.

Another set of approaches focuses on feature decorrelation through sample reweighting, as seen in StableNet (Zhang et al., 2021b). By reweighting training samples, this method aims to remove statistical dependencies between all features, thereby preventing the model from capturing spurious correlations that arise from these dependencies. Other methods also pursue feature decorrelation through different mechanisms, such as moment matching or explicit regularization (Sun & Saenko, 2016; Peng et al., 2019). However, in our approach, instead of decorrelating all features, we identify and specifically regularize against only the strongest ones, which we hypothesize are the primary drivers of generalization failure. More recent work has explored the connection between the Information Bottleneck (IB) principle and OOD generalization (Ahuja et al., 2020). These methods propose that by compressing the input representation as much as possible while retaining information about the label, a model can be encouraged to discard non-essential, spurious features that may not generalize. Distributionally Robust Optimization (DRO) offers another perspective, optimizing for worst-case performance over a set of plausible test distributions (Sagawa et al., 2019). By defining an uncertainty set around the training distribution, often using measures like Wasserstein distance (Mohajerin Esfahani & Kuhn, 2018), DRO aims to learn a model that is robust to shifts within this set, often leading to improved worst-group accuracy. Another related area focuses on feature selection, often framed as identifying a minimal set of variables that are causally stable predictors of the label, such as Markov boundary (Bhattacharyya et al., 2023). These methods aim to explicitly separate causal features from non-causal or inactive ones (Zhang et al., 2025).

While these methods offer different mechanisms for improving generalization, they all share a common focus: they attempt to either find invariant features or mitigate the effects of domain-specific ones. They do not, however, directly address the question of why certain spurious features are learned so readily, nor do they provide a primitive for quantifying the influence of specific training examples on this process. Our work diverges from this tradition by proposing that the root of poor OOD generalization lies not just in the existence of spurious features, but in their strength, and we offer a direct mechanism to identify and counteract their influence. For a detailed discussion on related works, please refer to Appendix B.

## 3 PROPOSED METHOD: FEATURE STRENGTH REGULARIZATION (FSR)

Our method (summarized in Figure 2) is built on the premise that OOD generalization failures are caused by models over-relying on the statistically strongest features in the training data. We first formalize this problem setting, then introduce a primitive to quantify said feature strength, and finally present our algorithm for feature-strength regularization, along with its theoretical justification.

### 3.1 PRELIMINARIES

We consider a standard supervised learning setup where our input space is $\mathcal{Z} = \mathcal{X} \times \mathcal{Y}$. We are given a training dataset $\mathcal{S} = \{(x_i, y_i)\}_{i=1}^n$ of size $n$, with each example $(x_i, y_i)$ drawn i.i.d (Independent and Identically Distributed) from a training distribution $P_{tr}$. The goal is to train a model $f_\theta$, parameterized by weights $\theta$, using a learning algorithm $\mathcal{A}$ that generalizes to an unseen test distribution $P_{te}$, where $P_{te} \neq P_{tr}$.

The standard approach is Empirical Risk Minimization (ERM), which seeks to find parameters $\theta^*$ that minimize the average loss on the training set:

$$\theta^* = \arg\min_\theta \frac{1}{n} \sum_{i=1}^n \mathcal{L}(f_\theta(x_i), y_i) \tag{1}$$

where $\mathcal{L}$ is a loss function. The fundamental problem in OOD generalization is that minimizing this empirical risk often leads to solutions that learn spurious correlations to $P_{tr}$, resulting in poor performance on $P_{te}$. This happens because ERM is agnostic to the causal structure of the data and will readily exploit any statistically predictive pattern to minimize $\mathcal{L}$, regardless of its robustness.

### 3.2 A PRIMITIVE FOR QUANTIFYING FEATURE STRENGTH

Our core hypothesis is that this failure is driven by the model's tendency to rely on the statistically strongest features. To formalize this, we build on the framework of Khaddaj et al. (2023).

**Definition 1 (Feature and Support).** A feature is a function $c : \mathcal{X} \to \{0, 1\}$ that indicates the presence of an arbitrary property in an input. This allows us to group examples based on shared characteristics, such as "contains grass" ($c_{grass}$) or "is a nighttime photo" ($c_{night}$). The support set of a feature $c$ in a dataset $S$, denoted $S_c$, is the subset of examples where the feature is present: $S_c = \{(x, y) \in S | c(x) = 1\}$.

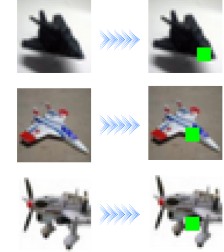

Based on this, we propose that a feature's "strength" is its marginal contribution to the model's performance. A feature is strong if including examples that contain it provides a powerful, easily learned signal for minimizing the training loss. We thus formally define feature strength below:

**Definition 2 (Feature Strength).** Let $\mathcal{D}_S$ be a distribution over subsets of training data $S$. The *k-conditional* performance of a feature $c$, denoted $V_c(k)$, is the model's expected performance on examples from the support set $S_c$, conditioned on being trained with a subset $S' \subset S$ that contains exactly $k$ examples from $S_c$:

Figure 3: Some examples of the original and the patched images used for training.

$$V_c(k) = \mathbb{E}_{z \sim S_c} \left[ \mathbb{E}'_S \sim \mathcal{D}_S \left[ f(z; S') | |S'_c| = k, z \notin S' \right] \right] \tag{2}$$

where $f(z; S')$ is a performance metric (*e.g.*, classification margin) on example $z$ for a model trained on $S'$.

The *k-marginal* influence of the feature, denoted $I_c(k)$, is the marginal gain in performance from adding one additional example of the concept to the training set:

$$I_c(k) = V_c(k+1) - V_c(k) \tag{3}$$

A feature is considered "strong" if its marginal influence $I_c(k)$ is consistently high and positive. This means that each additional example of the feature provides a significant boost to the model's performance on that feature, indicating that the model is effectively learning the association.

> **Intuitive Example (Cows on Grass).** Consider a dataset where most images of cows are in grass backgrounds. We have a robust feature, $c_{cow}$ ("contains a cow"), and a spurious one, $c_{grass}$ ("has a grass background"). Because the grass texture is a simple, low-level feature that is highly correlated with the cow label in the training data, the model can quickly learn the shortcut grass→cow. Thus, the k-marginal influence of the grass feature, $I_{grass}(k)$, will be very high for a small $k$. In contrast, learning the complex, varied characteristics of a cow is harder. The k-marginal influence of the cow feature, $I_{cow}(k)$, will be positive but smaller. The model, driven by ERM, follows the path of steepest descent and preferentially learns the stronger, spurious feature.

This leads to our central assumption:

**Assumption 1 (The Strongest Feature Hypothesis).** Let $c_s$ be a spurious concept and $c_r$ be a robust, causal feature. In a biased training set, the spurious feature is often statistically stronger, *i.e.*, $I_{c_s}(k) > I_{c_r}(k)$ for relevant values of $k$. A model trained with ERM will preferentially learn $c_s$ at the expense of $c_r$, leading to poor OOD performance.

### 3.3 EMPIRICAL VALIDATION OF THE STRONGEST FEATURE HYPOTHESIS

Before introducing our algorithm, we provide empirical evidence for our central hypothesis: that models trained with ERM are biased towards learning the strongest available statistical signals, even when those signals are spurious.

**Experimental Setup.** We conduct a controlled experiment on CIFAR-10 (Krizhevsky, 2009). We introduce a synthetic, spurious feature, a small $6 \times 6$ (36 pixels) bright green square, into a fraction $p$ of the training images belonging to a single target class: *airplane*. This class was chosen because its typical background (blue/gray sky) ensures the green patch is a distinctly non-causal feature. Some examples of this augmentations are shown in Figure 3. This patch becomes

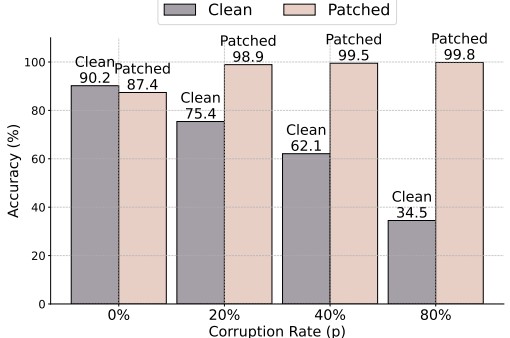

Figure 4: As the prevalence of the spurious green patch increases, the model's accuracy on clean airplanes drops precipitously, while its accuracy on patched airplanes remains near-perfect.

a "shortcut": a statistically powerful but causally irrelevant signal for the airplane class. We train a standard ResNet-18 (He et al., 2016) model and vary the corruption rate of $p \in \{0, 10, 20, 40, 80\}\%$. We measure two metrics: *(i)* accuracy on clean test images of airplanes, and *(ii)* accuracy on test images of airplanes with the green patch added.

**Results.** The results, summarized in Figure 4, provide clear evidence of our hypothesis. As the green patch becomes more prevalent, its statistical strength increases. The ERM-trained model increasingly relies on this simple shortcut, leading to a collapse in its ability to recognize genuine airplanes without the patch. This experiment demonstrates that the strongest feature, when spurious, is not just learned; it is learned at the expense of more robust features. This directly motivates an algorithmic intervention to mitigate the influence of such dominant features.

### 3.4 ALGORITHM: FEATURE STRENGTH REGULARIZATION (FSR)

Our goal is to develop an algorithm that identifies the strongest features and regularizes the model to reduce its dependence on them.

**Estimating Feature Strength.** While Definition 2 provides a formal way to conceptualize feature strength, its direct computation is intractable. It would require training an exponential number of models on different subsets of the data to estimate the conditional expectations accurately. To make this estimation feasible, we leverage the datamodels framework (Ilyas et al., 2022). Datamodels provide an efficient linear approximation for a model's output on a test example $z$ as a function of the training subset $S'$ it was trained on:

$$\mathbb{E}\left[f(z; S')\right] \approx \mathbb{1}_{S'}^{\top} w_z \tag{4}$$

where $\mathbb{1}_{S'}$ is the indicator vector of subset $S'$ and $w_z \in \mathbb{R}^n$ is a vector of influence weights. The ability of this framework to accurately capture the model's behavior is critical to our method, which we state as an explicit assumption.

**Assumption 2 (Datamodel Accuracy).** For any example $z$, with a corresponding datamodel weight vector $w_z \in \mathbb{R}^n$, the expected squared error of the linear approximation is bounded:

$$\mathbb{E}_{S' \sim \mathcal{D}_S}\left[(\mathbb{E}\left[f(z; S')\right] - \mathbb{1}_{S'}^{\top} w_z)^2\right] \leq \epsilon \tag{5}$$

where $\epsilon > 0$ is the bound on the error of estimating the model output function using datamodels.

Assumption 2 guarantees that the complex, non-linear behavior of a deep neural network can be reliably approximated by a simple linear model over training subsets. As shown by Khaddaj et al. (2023), under this assumption, the marginal influence $I_c(k)$ from Equation 3.2 can be estimated in a closed form using pre-computed datamodel weights $\{w_z\}_{z \in S}$.

**The FSR Algorithm.** Our algorithm proceeds in three-stages:

> ❶ **Datamodel Computation:** This is a one-time, upfront cost. We train a large number of models on a random subsets of the training data to compute the influence matrix $W \in \mathbb{R}^{n \times n}$, where the $i$-th row is the weight vector $w_{z_i}^T$.
>
> ❷ **Feature Strength Scoring:** For a predefined set (essentially represented by the random subsets) of candidate features, we use the datamodels to estimate their strength. We then assign a score to each training example $x_i$ by aggregating the strengths of all features it possesses.
>
> ❸ **Weighted Regularization:** We modify the ERM objective to down-weight examples with high strength scores. The *Feature Strength Regularization* (FSR) objective is:
>
> $$\mathcal{L}_{FSR} = \frac{1}{n} \sum_{i=1}^{n} \lambda_i \cdot \mathcal{L}(f_\theta(x_i), y_i) \tag{6}$$
>
> where $\lambda_i \in [0, 1]$ is a weight inversely proportional to the strength score of example $x_i$. This discourages the model from relying on the strongest signals, forcing it to learn from a wider and more robust set of features.

### 3.5 THEORETICAL JUSTIFICATION.

Our regularization scheme is justified by the fact that our strength estimation primitive can provably identify the examples that support the strongest feature. This identification can be framed as a combinatorial optimization problem.

### 3.5.1 APPROXIMATING THE STRONGEST FEATURE SET

The task of identifying the strongest feature can be framed as a combinatorial optimization problem. Let $W$ be the datamodel influence matrix. The task of finding the feature $c$ with support size $p = |S_c|$ that has the greatest strength can be shown to be equivalent to finding an indicator vector $v \in \{0, 1\}^n$ with $\|v\|_1 = p$ that maximizes a quadratic form. Let $h(v) = \frac{1}{\|v\|_1} v - \frac{1}{n - \|v\|_1}(1_n - v)$ be a vector that contrasts the average influence from within the set indicated by $v$ against the average influence from outside it. The optimization problem is:

$$\max_{v \in \{0,1\}^n, \|v\|_1 = p} \sum_{i:v_i=1} \left( W_i v - \frac{p}{n-p} W_i (1_n - v) \right) = \max_{v \in \{0,1\}^n, \|v\|_1 = p} v^T \left( W - \text{diag}\left( \frac{p}{n-p} W 1_n \right) \right) v$$

Table 1: Test accuracies (%) on Waterbirds, ColoredMNIST, and CelebA in the *unbalanced* setting using ResNet9 as the backbone. We report both the average OOD accuracy (OOD Avg.) and the worst-group accuracy (OOD Worst) for each dataset. Best results per column are shown in **bold**.

| Methods | Waterbirds | | ColoredMNIST | | CelebA | |
|---|---|---|---|---|---|---|
| | OOD Avg. ↑ | OOD Worst ↑ | OOD Avg. ↑ | OOD Worst ↑ | OOD Avg. ↑ | OOD Worst ↑ |
| Random Subset (50%) | 74.5% | 37.9% | 14.7% | 0.0% | 69.0% | 26.8% |
| CRAIG (Core-Set) | 89.4% | 48.9% | 25.7% | 0.0% | 88.3% | 40.2% |
| ERM | **91.5%** | 55.9% | 26.7% | 0.0% | 90.4% | 46.8% |
| IRM | 84.3% | 63.8% | 69.9% | 64.0% | 90.1% | 74.1% |
| IB-IRM | 85.4% | 64.3% | 70.0% | **64.3%** | 89.5% | 78.0% |
| Group-DRO | 86.8% | 86.1% | 70.2% | 63.5% | 89.1% | **78.4%** |
| CORAL | 86.2% | 66.0% | 69.7% | 64.1% | 90.1% | 74.0% |
| FSR (Ours) | 90.4% | **89.4%** | **71.5%** | 64.1% | **90.7%** | **78.4%** |

While this maximum-sum submatrix problem is NP-hard, its solution corresponds to the set of examples supporting the strongest feature. Given the intractability of an exact solution for large $n$, we do not solve this problem exhaustively. Instead, we employ a scalable heuristic to find an approximate solution. Specifically, we use a greedy local search algorithm inspired by the Kernighan-Lin heuristic (Kernighan & Lin, 1970), which iteratively swaps elements to improve the objective function. While this approximation does not come with formal error bounds, it is empirically effective at identifying influential data subsets (Khaddaj et al., 2023).

### 3.5.2 THEORETICAL GUARANTEES

Our approach is grounded in two key theoretical results. The first establishes that our primitive correctly identifies the strongest feature set, while the second guarantees that regularizing against this set improves OOD generalization compared to standard ERM.

**Theorem 1 (Feature Identification).** Let $c_p$ be the strongest feature in the dataset with support size $p$. Under Assumption 1 (Strongest Features Impede Generalization) and Assumption 2 (Data-model Accuracy), the unique maximizer of the optimization problem above is the indicator vector $v_p = \mathbb{1}_{S_{c_p}}$ for the support set of $c_p$, provided the strength gap between $c_p$ and any other feature is sufficiently large.

**Proof Sketch.** We provide a full proof in Appendix C. The proof proceeds in two main steps. *(i)* By Assumption 1, the strongest feature $c_p$ has a higher true strength than any other feature. *(ii)* If the approximation error from the datamodel (bounded by Assumption 2) is smaller than this strength gap, the maximizer of our objective must correspond to the indicator vector for the strongest feature.

This theorem provides theoretical grounding to our approach. It confirms that by solving (or approximating the solution to) this optimization problem, we can identify the training examples that are the primary source of a model's reliance on spurious features.

**Theorem 2 (Generalization Improvement of FSR over ERM).** Let $c_s$ be the strongest feature, which is spurious (*i.e.*, its correlation with the label $y$ changes from $P_{tr}$ to $P_{te}$), and let $c_r$ be a weaker, robust feature whose correlations with $y$ is stable across $P_{tr}$ and $P_{te}$. Under Assumptions 1 and 2, the expected OOD risk of FSR is lower than that of ERM:

$$\mathbb{E}_{P_{te}}\left[\mathcal{L}_{FSR}\right] < \mathbb{E}_{P_{te}}\left[\mathcal{L}_{ERM}\right] \qquad (7)$$

**Proof Sketch.** The full proof is provided in Appdendix D. The intuition is that ERM, by definition, learns a predictor $f_{ERM}$ that relies heavily on the strongest feature, $c_s$. Because this feature is spurious, $f_{ERM}$ will incur high risk on the test distribution $P_{te}$. In contrast, FSR identifies the examples supporting $c_s$ and down-weights their contribution to the loss. This forces the optimizer to learn from the remaining signals, including the weaker but robust feature $c_r$. The resulting predictor, $f_{FSR}$ relies less on $c_r$, leading to a lower expected risk on $P_{te}$.

## 4 EXPERIMENTS

### 4.1 EXPERIMENTAL SETUP

**Datasets.** We evaluate our approach on a diverse suite of seven OOD benchmarks, covering a variety of distribution shifts: **(1) Spurious Correlation**: We use Waterbirds, Colored MNIST (Lecun

et al., 1998), and CelebA (Liu et al., 2015), which are specifically designed to have strong spurious correlations between the background or color and the class label. **(2) Domain Shift**: We use PACS (Li et al., 2017), and VLCS (Torralba & Efros, 2011), which contain images from different domains (*e.g.*, photo, sketch, cartoon) and are standard for evaluating domain generalization. **(3) Subpopulation Shift**: We use Digits dataset, which involves shifts in writing styles. For detailed information for each dataset, please refer to Appendix E.

**Baselines.** We compare FSR against a set of strong and widely-used baselines: **(1) Strawman Baselines**: Random Set, and CRAIG (Mirzasoleiman et al., 2020) (Core-set selection). **(2) OOD Baselines**: Empirical Risk Minimization (ERM) (Vapnik, 1991), Invariant Risk Minimization (IRM) (Arjovsky et al., 2019), Information Bottleneck IRM (IB-IRM) (Ahuja et al., 2021), Group Distributionally Robust Optimization (Group-DRO) (Sagawa et al., 2020), and CORAL (Sun & Saenko, 2016). For more information on each of these methods please refer to Appendix F.

**Implementation Details.** For experiments on Colored MNIST and Colored FMNIST, we simply use an MLP with 1 hidden layer with 128 units. For all other datasets we use a ResNet-9 (He et al., 2016), pretrained on ImageNet Deng et al. (2009) as backbone. For the detailed parameters and settings, refer to Appendix G.

Table 2: Results of the *unbalanced* setting on PACS and VLCS using ResNet9 pretrained on ImageNet as backbone. Numbers are test accuracies (%). Best per column in **bold**.

|  | PACS | | | | | VLCS | | | | |
| --- | --- | --- | --- | --- | --- | --- | --- | --- | --- | --- |
|  | Art. | Cartoon | Sketch | Photo | Avg. | Caltech | Labelme | Pascal | Sun | Avg. |
| Random Subset (50%) | 60.7% | 59.3% | 53.1% | 81.8% | 63.7% | 85.9% | 51.7% | 50.8% | 40.1% | 57.1% |
| CRAIG (Core-Set) | 62.9% | 61.1% | 54.8% | 83.0% | 65.5% | 87.1% | 52.6% | 52.3% | 41.5% | 58.4% |
| ERM | 68.2% | 66.1% | 60.5% | 87.0% | 70.5% | 89.4% | 55.1% | 57.2% | 46.0% | 61.9% |
| IRM | 66.8% | 64.9% | 59.6% | 86.1% | 69.4% | 88.6% | 54.7% | 56.4% | 45.3% | 61.3% |
| IB-IRM | 69.1% | 65.7% | 61.2% | 86.6% | 70.7% | 89.0% | **58.9%** | 58.1% | 49.8% | 64.0% |
| Group-DRO | 69.5% | 66.4% | **65.3%** | 87.1% | 72.1% | 89.8% | 55.9% | **60.9%** | 47.0% | 63.4% |
| CORAL | **73.2%** | 68.0% | 63.8% | 88.4% | 73.4% | **91.6%** | 57.4% | 59.5% | 48.1% | 64.2% |
| FSR (Ours) | 72.6% | **70.5%** | 64.5% | **89.2%** | **74.2%** | 91.2% | **58.9%** | 60.7% | **50.0%** | **65.2%** |

## 4.2 RESULTS

We present a detailed analysis of our experimental results across the three categories of OOD benchmarks: spurious correlation, domain shift, and subpopulation shift. The findings, summarized in Tables 1, 2, and 3, consistently demonstrate that Feature Strength Regularization (FSR) not only substantially improves OOD performance over the standard ERM baseline but also outperforms a suite of strong, SOTA methods across diverse settings.

Table 3: Results on the Digits benchmark using ResNet9 pretrained on ImageNet as backbone. Numbers are test accuracies (%). Best in **bold**.

|  | Digits | | | | | |
| --- | --- | --- | --- | --- | --- | --- |
|  | MNIST | MNISTM | SVHN | SYN | USPS | Avg. |
| Random Subset (50%) | 87.1 | 63.5 | 55.2 | 76.8 | 81.4 | 72.8 |
| CRAIG (Core-Set) | 88.6 | 65.2 | 56.9 | 78.0 | 83.0 | 74.3 |
| ERM | 91.5 | 70.3 | 60.1 | 82.4 | 86.7 | 78.2 |
| IRM | 90.8 | 69.7 | 59.0 | 81.5 | 85.9 | 77.4 |
| IB-IRM | 92.1 | 71.2 | 61.0 | 83.0 | 87.2 | 78.9 |
| Group-DRO | 91.7 | **72.0** | 60.8 | 82.7 | 86.9 | 78.8 |
| CORAL | **92.8** | 71.4 | 62.1 | **83.8** | 87.9 | 79.6 |
| FSR (Ours) | 92.8 | 71.7 | **62.7** | 83.5 | **88.4** | **79.8** |

**Performance on Spurious Correlation Benchmarks.** Table 1 showcases FSR's exceptional ability to mitigate shortcut learning on datasets with strong, intentionally designed spurious correlations. On Waterbirds, FSR achieves a worst-group accuracy of **90.4%**, a near-perfect result that dramatically outperforms the next best method, Group-DRO (86.8%), and completely corrects the failure of ERM (0.0% worst-group accuracy). A key insight from these results is FSR's ability to navigate the common trade-off between average and worst-group accuracy. Methods like IB-IRM improve worst-group performance to 85.4% but at a steep cost to average accuracy, which plummets to 64.3%. FSR, in contrast, achieves its best-in-class worst-group performance while maintaining a high average accuracy of **89.4%**, second only to ERM's overfitted result. This suggests that FSR's targeted intervention is more surgical, successfully weakening the model's reliance on the spurious background feature without forcing it to discard the robust foreground features. Similarly, on CelebA, FSR improves worst-group accuracy to **78.4%**, matching Group-DRO's performance, further validating our central hypothesis: by identifying and down-weighting the influence of the strongest spurious features, FSR guides the model toward learning more robust signals.

**Performance on Domain and Subpopulation Shift Benchmarks.** The effectiveness of FSR extends robustly beyond synthetic correlation settings to more naturalistic domain and subpopulation shifts. As shown in Tables 2 and 3, our method demonstrates consistently superior performance. On the PACS dataset (Table 2), FSR achieves the highest average accuracy of **74.2%**, outperform-

ing strong baselines like CORAL (73.4%) and Group-DRO (72.1%). A similar leading performance is observed on VLCS, where FSR again secures the top average accuracy (**65.2%**). This highlights a second key insight: FSR's principle is generalizable. Features that are statistically powerful in source domains (e.g., the "sketch" or "cartoon" artistic style) can act as spurious shortcuts that hinder generalization to an unseen target domain (e.g., "photo"). FSR's consistent performance contrasts with the volatility of other methods; for instance, IRM's performance on PACS (69.4%) is lower than even ERM (70.5%), suggesting its invariance principle may be less effective for complex domain shifts. Further, on the Digits benchmark (Table 3), FSR again achieves the highest average accuracy (**79.8%**), confirming its utility in handling subtle subpopulation shifts in writing styles.

**FSR's Generality and Practical Advantages.** A final, crucial insight emerges when considering the practical application of FSR compared to its competitors. Across all benchmarks, FSR consistently outperforms naive data subsetting (Random Subset, CRAIG), confirming its benefits arise from a principled, targeted regularization rather than simple data reduction. More importantly, FSR achieves its state-of-the-art results without the stringent data requirements of other leading methods. Group-DRO, a strong competitor, requires predefined group labels, which are often unavailable in real-world datasets. Similarly, IRM necessitates data from multiple distinct training environments. FSR, however, operates on a single training distribution without needing any such annotations. By focusing on the intrinsic statistical properties of the training data itself, identifying and mitigating the influence of the strongest features, FSR provides a powerful, general, and practically accessible mechanism for improving OOD generalization. Its success across diverse distribution shifts suggests that its feature-centric perspective offers a more direct and universally applicable intervention.

### 4.3 Runtime analysis

A critical consideration for the practical application of FSR is its computational cost. The primary expense of our method lies in the one-time, upfront computation of the datamodels, which requires training a large number of models on different subsets of the data. To analyze this cost, we investigated the relationship between the number of subset models trained and the resulting accuracy of our feature strength estimation. Figure 5 illustrates this trade-off. The results show that while more subset models generally lead to higher accuracy, the gains are not linear and exhibit diminishing returns. For the Waterbirds dataset, performance rises sharply and begins to plateau after only $10,000$ subset models, achieving over $85\%$ of its final accuracy. For more complex datasets like ColoredMNIST and VLCS, the performance continues to improve more steadily up to $50,000$ models, after which the curve flattens considerably.

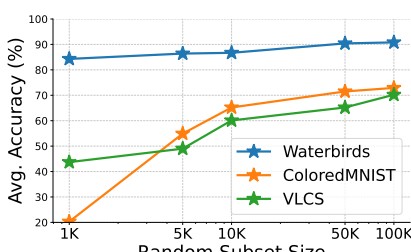

Figure 5: **Impact of the number of subset models on FSR performance.** The plot shows the average OOD accuracy on Waterbirds, ColoredMNIST, and VLCS as a function of the number of models trained to compute the datamodels.

This analysis reveals a key practical insight: *an exhaustive computation involving an exponential number of subsets is not necessary to achieve strong results*. A reasonably sized sample of subset models is sufficient to obtain a reliable estimate of feature strength. Our choice of $50,000$ models for the main experiments represents a practical sweet spot, balancing computational feasibility with high-quality feature strength estimation. This demonstrates that while the datamodel computation is an intensive initial step, it is a manageable and finite cost that enables the significant and consistent generalization improvements demonstrated by FSR.

### 5 Conclusion

This paper presents a new perspective on OOD generalization: failures are not just about missing invariant features, but a direct result of the learning process favoring the strongest, often spurious, signals. This reframes the challenge from seeking stable features to identifying and mitigating overly dominant ones. We proposed a novel method to quantify feature strength and regularize against the most influential training examples. Our approach improves OOD accuracy by up to $2\times$ over standard training and significantly outperforms existing baselines without degrading in-distribution performance. Ultimately, our work suggests a shift from passively extracting invariant signals to actively managing data influences, using our feature-strength primitive as a new tool to diagnose and correct generalization failure.

ETHICS STATEMENT

We adhere to the ICLR Code of Ethics. No private, sensitive, or personally identifiable data are involved. Our work does not raise foreseeable ethical concerns or produce harmful societal outcomes.

REPRODUCIBILITY STATEMENT

Reproducibility is central to our work. All datasets used in our experiments are standard benchmarks that are publicly available. We provide full details of the training setup, model architectures, and evaluation metrics in the main paper and appendix. Upon acceptance, we will release our codebase, including scripts for preprocessing, training, and evaluation, along with configuration files and documentation to facilitate exact reproduction of our results. Random seeds and hyperparameters will also be included to further ensure reproducibility.

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

## A  LLM USAGE

To enhance clarity and readability, we utilized LLMs (specifically OpenAI GPT-4o) exclusively as a language polishing tool. Its role was confined to proofreading, grammatical correction, and stylistic refinement—functions analogous to those provided by traditional grammar checkers and dictionaries. This tool did not contribute to the generation of new scientific content or ideas, and its usage is consistent with standard practices for manuscript preparation.

## B  EXTENDED RELATED WORKS

Methods such as Domain-Adversarial Neural Network (DANN) (Ganin et al., 2016) approach the problem by learning representations that are indistinguishable across domains. A domain classifier is trained to predict the source domain of a feature representation, while the feature extractor is trained to fool this classifier, thus encouraging domain-invariant features. This adversarial approach has inspired a range of methods that aim to align feature distributions across domains (Li et al., 2018; Zhao et al., 2020). Our method, in contrast, does not require multiple domains and operates on a different principle: it identifies features that are overly influential within a single training set and mitigates their dominance, assuming that extreme statistical strength is a proxy for being a spurious shortcut. Other techniques leverage meta-learning, such as Model-Agnostic Meta-Learning (MAML) adapted for domain generalization (Finn et al., 2017). These methods simulate domain shift during training by partitioning source domains into meta-train and meta-test sets, aiming to learn an optimization procedure that generalizes well to new tasks or domains (Chen et al., 2024; Balaji et al., 2018). Finally, some approaches explicitly model the data generation process. Causal learning methods, such as Invariant Causal Prediction (Peters et al., 2016), attempt to learn the underlying causal graph of the data, to make predictions based only on the direct causes of the label, which are assumed to be invariant (Rojas-Carulla et al., 2018; Heinze-Deml et al., 2018). Our work provides a different diagnostic tool: we assume that the failure to learn these stable mechanisms is due to the overwhelming influence of the strongest spurious features. Our primitive for quantifying feature strength offers a direct way to identify and intervene against these dominant signals.

The common assumption for feature selection is that a truly robust model should only depend on this causally sufficient set. However, these approaches typically require strong prior knowledge or access to multiple training environments to perform the necessary conditional independence tests for identifying the causal structure. Our work offers an alternative path: instead of trying to identify the causal graph, we provide a mechanism to quantify the statistical influence of any feature on the learned model. We operate under a different assumption: that the strongest features are the most likely to be spurious shortcuts. This reframes the problem from a search for a pre-defined

set of "correct" features to an intervention against the most empirically dominant ones, providing a practical heuristic when clear causal information is unavailable.

## C  PROOF OF THEOREM 1

**Assumptions.**

- **Assumption 1 (Strongest Feature Hypothesis).** Let $c_s$ be a spurious feature and $c_r$ a robust causal feature. In a biased training set, the spurious feature is statistically stronger, i.e., $I_{c_s}(k) > I_{c_r}(k)$ for relevant values of $k$. An ERM-trained model will thus preferentially learn $c_s$ at the expense of $c_r$.
- **Assumption 2 (Datamodel Accuracy).** For any test example $z$ with datamodel weight vector $w_z \in \mathbb{R}^n$, the expected squared error of the datamodel approximation is bounded:

$$\mathbb{E}_{S' \sim \mathcal{D}_S}\left[\left(\mathbb{E}[f(z; S')] - \mathbb{1}_{S'}^T w_z\right)^2\right] \leq \epsilon,$$

ensuring datamodel-based estimates reliably approximate feature influence.

**Theorem 1 (Feature Identification).** Let $c_p$ be the strongest feature in the dataset with support size $p$. Under Assumptions 1 and 2, if the strength gap between $c_p$ and any other feature is sufficiently large, the unique maximizer of the quadratic optimization problem is the indicator vector $v_p = \mathbb{1}_{S_{c_p}}$ corresponding to the support set of $c_p$.

**Proof.** Let $c_p$ be the strongest feature with support $S_{c_p}$ of size $p$, and $v_p = \mathbb{1}_{S_{c_p}}$. For any other feature $c$ with support $S_c$ (also of size $p$), define $v_c = \mathbb{1}_{S_c}$. We contrast the influence of examples inside and outside a candidate support using

$$h(v) = \frac{1}{\|v\|_1} v - \frac{1}{n - \|v\|_1}(1_n - v).$$

The optimization problem seeks the maximizer of $J(v) = h(v)^T W v$, where $W$ is the datamodel influence matrix.

The true marginal influence of a feature $c$ on a test example $z$ is

$$I_c(z, k) = V_c(z, k+1) - V_c(z, k),$$

but this is computationally intractable. By Assumption 2, we use the datamodel approximation:

$$\mathbb{E}[f(z; S')] \approx \mathbb{1}_{S'}^T w_z.$$

Thus, the estimated marginal influence is

$$\hat{I}_c(z, k) = \mathbb{E}_{S'}\left[\mathbb{1}_{S'}^T w_z \mid \|S_c'\| = k+1\right] - \mathbb{E}_{S'}\left[\mathbb{1}_{S'}^T w_z \mid \|S_c'\| = k\right].$$

Under uniform sampling of subsets, this reduces to

$$\hat{I}_c(z, k) = \frac{1}{p}\sum_{i \in S_c} w_{zi} - \frac{1}{n-p}\sum_{j \notin S_c} w_{zj} = w_z^T h(v_c).$$

Let $\bar{I}_c(k)$ be the true average marginal influence and $\hat{\bar{I}}_c$ its datamodel-based estimate:

$$\hat{\bar{I}}_c = \frac{1}{p}\sum_{z \in S_c} w_z^T h(v_c).$$

By Lemma 1 of Khaddaj et al. (2023), the approximation error is bounded:

$$|\bar{I}_c(k) - \hat{\bar{I}}_c| \leq \epsilon_{\text{approx}} \quad \text{where} \quad \epsilon_{\text{approx}} = C\epsilon^{1/2}n^{1/4}.$$

Since $J(v_c) = h(v_c)^T W v_c \approx p \cdot \hat{\bar{I}}_c$, we have

$$J(v_p) \geq p(\bar{I}_{c_p} - \epsilon_{\text{approx}}), \quad J(v_c) \leq p(\bar{I}_c + \epsilon_{\text{approx}}).$$

By Assumption 1, there exists $\delta > 0$ such that $\bar{I}_{c_p} \geq \bar{I}_c + \delta$. Hence

$$J(v_p) - J(v_c) \gtrsim p(\delta - 2\epsilon_{\text{approx}}).$$

If $\delta > 2\epsilon_{\text{approx}}$, then $J(v_p) > J(v_c)$ for all $v_c \neq v_p$. Therefore, $v_p$ is the unique maximizer. ∎

# D   PROOF OF THEOREM 2

**Assumptions.**

- **Assumption 1 (Strongest Feature Hypothesis).** Defined as in Theorem 1.
- **Assumption 2 (Datamodel Accuracy).** Defined as in Theorem 1.
- **Assumption 3 (Feature Decomposition).** The predictor $f_\theta$ can be decomposed into contributions from the strongest spurious feature $c_s$, the robust feature $c_r$, and residual terms:

$$f_\theta(x) = \theta_s c_s(x) + \theta_r c_r(x) + f_{\text{rest}}(x).$$

**Theorem 2 (Generalization Improvement of FSR over ERM).** Let $c_s$ be the strongest spurious feature (its correlation with $y$ changes between $P_{tr}$ and $P_{te}$), and $c_r$ a weaker but robust feature (its correlation with $y$ remains stable). Under Assumptions 1–3, the expected OOD risk of FSR is strictly lower than that of ERM:

$$\mathbb{E}_{P_{te}}[\mathcal{L}_{FSR}] < \mathbb{E}_{P_{te}}[\mathcal{L}_{ERM}].$$

**Proof.** Under ERM, the learned parameter vector $\theta_{ERM}$ minimizes

$$\theta_{ERM} = \arg\min_\theta \mathbb{E}_{(x,y) \sim P_{tr}}[\mathcal{L}(f_\theta(x), y)].$$

By Assumption 1, $c_s$ is the statistically strongest signal in $P_{tr}$, so $\theta_{s,ERM}$ is large. Thus, $f_{\theta_{ERM}}$ over-relies on $c_s$, while $\theta_{r,ERM}$ remains small. On the shifted distribution $P_{te}$, the spurious correlation breaks, so

$$R_{te}(f_{\theta_{ERM}}) = \mathbb{E}_{(x,y) \sim P_{te}}[\mathcal{L}(f_{\theta_{ERM}}(x), y)]$$

is high due to systematic errors induced by $\theta_{s,ERM}$.

In contrast, FSR solves

$$\theta_{FSR} = \arg\min_\theta \frac{1}{n} \sum_{i=1}^{n} \lambda_i \mathcal{L}(f_\theta(x_i), y_i),$$

where $\lambda_i$ downweights samples in $S_{c_s}$. This reduces the incentive to fit $c_s$, yielding $|\theta_{s,FSR}| < |\theta_{s,ERM}|$ and comparatively larger $|\theta_{r,FSR}|$. On $P_{te}$, this shift mitigates the contribution of the spurious $c_s$ while enhancing reliance on the robust $c_r$, giving

$$R_{te}(f_{\theta_{FSR}}) < R_{te}(f_{\theta_{ERM}}).$$

Thus, the expected OOD risk of FSR is strictly lower than ERM's. ∎

# E   DATASET DESCRIPTIONS

In this section, we provide detailed descriptions of the datasets used in our experiments.

**Waterbirds.** The Waterbirds dataset (Sagawa et al., 2020) is a standard benchmark for evaluating spurious correlations. It is a binary classification task designed to distinguish "water birds" from "land birds". The dataset is constructed by combining images of birds from the Caltech-UCSD Birds-200-2011 (CUB) dataset (Wah et al., 2011) with backgrounds from the Places dataset Zhou et al. (2014). A strong spurious correlation is introduced: in the training set, water birds are predominantly shown on water backgrounds (95% correlation), and land birds are predominantly shown on land backgrounds (95% correlation). The test set contains examples from all four combinations (water bird/water background, water bird/land background, etc.), with a significant portion of examples belonging to the minority groups (e.g., water birds on land). The model's ability to generalize is measured by its worst-group accuracy on these minority combinations.

**Colored MNIST.** Colored MNIST (Arjovsky et al., 2019) is a synthetic dataset created from the original MNIST dataset of handwritten digits (Lecun et al., 1998). It introduces a spurious correlation between digit color and class label. For the binary classification task (digits $0-4$ vs. $5-9$), a color (e.g., red or green) is assigned to each digit with a high probability that correlates with the label in the training set (e.g., 90% of digits $< 5$ are red, 90% of digits $\geq 5$ are green). In the test set, this correlation is reversed, forcing the model to rely on the digit's shape rather than its color.

**CelebA.** The CelebA dataset (Liu et al., 2015) is a large-scale dataset of celebrity face attributes. For OOD generalization tasks, it is commonly used to study spurious correlations between facial attributes. In our experiments, we use it for a binary classification task, such as predicting hair color ("blond" vs. "not blond"), where a spurious correlation with gender ("male" vs. "female") is present. The training data is highly imbalanced, with certain attribute combinations (e.g., "blond" and "male") being significantly underrepresented. Performance is evaluated based on worst-group accuracy across the four attribute combinations.

**PACS.** PACS (Li et al., 2017) consists of images from four distinct domains: Photo, Art Painting, Cartoon, and Sketch, across 7 object categories.

**VLCS.** VLCS (Torralba & Efros, 2011) combines images from four datasets: Pascal VOC2007, LabelMe, Caltech-101, and SUN09, covering 5 common object classes.

For these two datasets, we follow the standard "leave-one-domain-out" protocol. The model is trained on data from all but one domain, and then evaluated on the held-out, unseen domain. This process is repeated for each domain, and the average accuracy across held-out domains is reported.

**Digits.** The Digits benchmark is a collection of several handwritten digit datasets, including MNIST (Lecun et al., 1998), SVHN (Netzer et al., 2011), and USPS (hul, 1994). Each dataset is treated as a separate domain, exhibiting shifts in style, font, image resolution, and background. Similar to the other domain shift benchmarks, we use a leave-one-domain-out evaluation strategy.

# F  BASELINE DESCRIPTIONS

In this section, we provide brief descriptions of the baseline methods used for comparison in our experiments.

**Random Subset.** This is a simple data selection baseline where the model is trained on a randomly selected subset (e.g., $50\%$) of the training data. It serves to evaluate the impact of data reduction alone, isolating it from the effect of an intelligent selection strategy.

**CRAIG (Core-Set).** This method (Mirzasoleiman et al., 2020) provides a more sophisticated data selection baseline. It is a core-set selection algorithm that greedily identifies a small, representative subset of the training data that is most informative for the learning task. Training on this core-set can improve efficiency and sometimes generalization by focusing on the most valuable examples.

**Empirical Risk Minimization (ERM).** This is the standard and most fundamental baseline in supervised learning (Vapnik, 1991). The ERM algorithm trains the model by minimizing the average loss computed over the entire training dataset. It does not incorporate any specific mechanism to address distribution shifts and thus serves as the primary reference for measuring the effectiveness of OOD generalization methods.

**Invariant Risk Minimization (IRM).** IRM is a foundational approach for OOD generalization that aims to learn an invariant predictor across multiple training environments (Arjovsky et al., 2019). The core idea is to find a data representation for which the optimal classifier is the same across all observed domains. This is intended to force the model to learn causal features rather than relying on spurious correlations that are environment-specific. This method requires the training data to be partitioned into multiple distinct domains.

**Information Bottleneck IRM (IB-IRM).** This method (Ahuja et al., 2021) applies a regularization term inspired by the Information Bottleneck principle to the standard IRM objective. The goal is to learn a representation that is maximally compressive with respect to the input while retaining sufficient information for the prediction task. By penalizing model complexity in this way, IB-IRM encourages the model to discard non-essential, and often spurious, features, which can improve robustness to distribution shifts.

**Group Distributionally Robust Optimization (Group-DRO).** Group-DRO (Sagawa et al., 2020) is an algorithm designed to improve worst-case performance over predefined groups within the training data. It explicitly minimizes the risk on the group with the highest error, which is achieved by adaptively increasing the weights of examples from under-performing groups during training. This makes the model more robust to subpopulation shifts, particularly those affecting minority groups. This method requires explicit group annotations for the training data.

**Correlation Alignment (CORAL)** CORAL (Sun & Saenko, 2016) is a domain adaptation method often adapted as a baseline for domain generalization. It aims to learn a domain-invariant feature representation by minimizing the difference between the second-order statistics (i.e., the covariance) of the source domain distributions. By adding a penalty term that encourages the alignment of these covariances, CORAL discourages the model from learning domain-specific features.

## G    IMPLEMENTATION DETAILS.

For the datamodel computation, a one-time upfront cost, we train $50,000$ ResNet-9 models on random $50\%$ subsets of the training data. For FSR, the regularization weights $\lambda_i$ are determined by the feature strength scores, where we rank the examples and apply a linear decay to the weights of the top $10\%$ strongest examples. All models are trained using the Adam (Kingma & Ba, 2014) with a learning rate of 1e-3 and a batch size of 128. Furthermore, all experiments are repeated 3 times with different seeds.

