# OpenReview forum: "Beyond Invariance: A Feature-Strength Framework for OOD Generalization"
_ICLR.cc/2026/Conference — ICLR 2026 Conference Withdrawn Submission_

### Official Review · Reviewer_RnsM · 2025-10-28

**Soundness:** 3
**Presentation:** 3
**Contribution:** 2
**Rating:** 4
**Confidence:** 5

**Summary:**

This paper introduces a feature-strength framework for OOD generalization. In particular, the paper argues that ERM-trained neural networks fail to generalize OOD due to an overly focus on dominant features, quantified by the "strength" of the feature. The paper then proposes a regularization method to weaken the model's reliance on dominant features, thus increasing the overall feature diversity. The proposed method is evaluated on a range of OOD generalization benchmarks.

**Strengths:**

The proposed quantification scheme of feature strength is of interest in the OOD generalization context.

**Weaknesses:**

The paper claims that the perspective that ERM fails to generalize OOD due to favoring the strongest feature is new. However, there is already prior work that introduces similar observations and/or analysis, e.g., [1, 2], to name a few. There are also targeted methods for biasing ERM to learn richer or more diverse features, e.g, through weight averaging [3, 4] or penalizing the effect of dominant features [5, 6], which shares nearly the same spirit as this paper.

The main ingredients of the proposed feature-strength framework (Sec. 3.2 and 3.5.1) are taken from Khaddaj et al. Overall, the technical contribution of the paper is limited.

The proposed method is only evaluated against a limited set of baselines. For example, weight averaging methods should be included on PACS and VLCS, and there should be more recent baselines on Waterbirds and CelebA.

Finally, the paper is written in a narrative that existing OOD generalization research mostly focuses on learning invariant representations. While invariance is indeed a prevailing perspective on OOD generalization, there are also analyses attributing the failure of OOD to limited feature diversity (e.g., [1-6]) or the inductive bias of the model (e.g, [7-9]). The authors are encouraged to add them to the related work section to avoid misunderstanding for future readers.

---

[1] The pitfalls of simplicity bias in neural networks. NeurIPS, 2020.

[2] Evading the simplicity bias: Training a diverse set of models discovers solutions with superior OOD generalization. CVPR, 2022.

[3] Spurious feature diversification improves out-of-distribution generalization. ICLR, 2024.

[4] Diverse weight averaging for out-of-distribution generalization. NeurIPS, 2022.

[5] Gradient starvation: A learning proclivity in neural networks. NeurIPS, 2021.

[6] Rich feature construction for the optimization-generalization dilemma. ICML, 2022.

[7] Understanding the failure modes of out-of-distribution generalization. ICLR, 2021.

[8] Feature contamination: Neural networks learn uncorrelated features and fail to generalize. ICML, 2024.

[9] Understanding and improving feature learning for out-of-distribution generalization. NeurIPS, 2023.

**Questions:**

See Weaknesses.

---

### Official Review · Reviewer_JckS · 2025-10-31

**Soundness:** 3
**Presentation:** 3
**Contribution:** 2
**Rating:** 2
**Confidence:** 4

**Summary:**

This paper presents a new perspective on Out-of-Distribution (OOD) generalization, arguing that model failures are not just a matter of missing invariant features, but a direct result of the learning algorithm, such as Empirical Risk Minimization (ERM), preferentially learning the statistically strongest features, which are often spurious shortcuts. Based on a few assumptions made, the authors prove that their algorithm can achieve smaller loss than the ERM algorithm. The authors run experiments on standard distribution shifts datasets to show that their method has better performance, albeit that the baselines are mostly before 2021.

**Strengths:**

Overall, this paper is well motivated and theoretically grounded. The study of OOD generalization is long-lasting, and there are a lot of empirical and theoretical stuides trying to formulate the problem from the aspects of feature properties, and regularizing the strengthes of features is relatively novel.

**Weaknesses:**

As a theory-motivated paper, I think the following issues are blocking the paper to reach the accpetance threshold. Specifically,

- Assumptions not carefully verified, and the empirical validation does not match the assumptions. Based my understanding, Assumption 1 is probably the most important assumption that summarizes the authors' intuition, and it is not a obvious thing. However, the authors fail to empirically demonstrate that the assumptions hold (even to some extent). The study in section 3.3 is quite different from what I think, as (1) the spurious feature is completely synthetic and could not reflect the practical scenarios, and (2) the relation $I_{c_s}(k) > I_{c_r}(k)$ is not analytically verified even in this synthetic scenario. Since Assumption 1 is the foundation of the proposed algorithm, it should be carefully verified to support the paper.
- Theoretical results not quantiative. I am not satisfied with the two theorems presented in Section 3. With all these assumptions and theoretical framing, I am surprised that the authors cannot present a quantiative improvement result compared with ERM. Theorem 2 is too weak as it give no implication on how much the proposed algorithm can outperform ERM, not to say the other OOD algorithms. In addition, the authors should also compare with other OOD analysis works, including "In Search of Lost Domain Generalization" as well as "Towards a Theoretical Framework of Out-of-Distribution Generalization".
- Experiments insufficiency. Overall, baselines compared in this paper is too outdated, with the latest one from 2021. This completely ignores the recent progress in the OOD generalization domains, and the authors should find at least 2 more recent baselines to compare. In addition, the authors are encouraged to evaluate their methods on the WILDS dataset, which is a magnitude larger than the existing datasets and can reflect their method's performance better.

**Questions:**

- Line 213: there could be typos in the formula. $S' \sim \mathcal D_S$ should be under the expectation.
-  Can you give a more intuitive explanation why the assumption 2 hold? I can understand the intuition behind the assumption 1 but for this one it's a bit hard to understand.

---

### Official Review · Reviewer_aAc2 · 2025-11-01

**Soundness:** 2
**Presentation:** 3
**Contribution:** 2
**Rating:** 2
**Confidence:** 3

**Summary:**

The paper suggests that the strongest features learned by ERM are likely to be spurious and less generalizable. To reduce such reliance, Feature Strength Regularization (FSR) is proposed. FSR calculates the strength of each feature using datamodels, and inversely weights each sample based on the summation of the strengths of the features it contains. Theoretical analysis shows that FSR can identify features with a strong influence and provides a better performance guarantee compared to ERM. Experiments on several datasets show the potential of FSR.

**Strengths:**

1. Using influence weight to measure strong and weak features is novel.

2. Theoretical analysis gives mathematical insight.

3. The paper is clearly written and easy to follow.

**Weaknesses:**

1. There are several existing works (e.g. RSC [1], MaskTune [2], FeAT [3]) that recognize the spurious features are likely to be dominant, and propose methods to reduce a model’s reliance on the strongest features to achieve better generalization. They should be mentioned and compared to show the advances of FSR.

2. FSR relies on a predefined set of features to capture the feature strength. To calculate $\lambda$ for each sample, an annotation of the features it contains needs to be available, which requires extra information in addition to the class labels. This hinders the generalizability of FSR.

3. Using datamodels to calculate feature strengths incurs very high computational cost, as thousands of extra models need to be trained. This will significantly reduce the scalability of FSR when larger architectures are required for complex datasets.

[1] Huang et al., Self-Challenging Improves Cross-Domain Generalization, ECCV 2020

[2] Taghanaki et al., MaskTune: Mitigating Spurious Correlations by Forcing to Explore, NeurIPS 2022

[3] Chen et al., Understanding and Improving Feature Learning for Out-of-Distribution Generalization, NeurIPS 2023.

**Questions:**

1. How is the predefined set of features determined for each dataset used in the experiment? How is the feature information collected for each sample? Given a new dataset, how should the features be defined to apply FSR?

---

### Official Review · Reviewer_AD82 · 2025-11-01

**Soundness:** 3
**Presentation:** 3
**Contribution:** 2
**Rating:** 2
**Confidence:** 4

**Summary:**

This work studies the problem of OOD generalization. The authors reframe the problem of OOD generalization as one of detecting and mitigating the influence of the strongest features in a dataset and propose Feature Strength Regularization (FSR) to avoid learning dominant features. Experiments on some OOD benchmarks show the effectiveness of FSR.

**Strengths:**

1. The feature learning perspective and the reframing of the OOD generalization problem are interesting and sound.

2. The presentation is clear and easy to follow.

3. Empirical results show improvements on some OOD benchmarks.

**Weaknesses:**

1. The primary problem has already been well-studied in the literature via the notion of rich feature learning[1,2,3,4]:

[1] Rich Feature Construction for the Optimization-Generalization Dilemma, ICML'22.

[2] Pareto Invariant Risk Minimization, ICLR'23.

[3] Understanding and Improving Feature Learning for Out-of-Distribution Generalization, NeurIPS'23.

[4] Learning useful representations for shifting tasks and distributions, ICML'23.


2. The method relies on the fitting of the data model, which may not be practical and require a sufficient number of samples.

3. Lack of comparison with the state-of-the-art baselines as well as real-world OOD benchmarks such as those in the Wlids benchmark.

**Questions:**

Please find the details in the section above.

---

### Official Review · Reviewer_mN5m · 2025-11-01

**Soundness:** 2
**Presentation:** 3
**Contribution:** 1
**Rating:** 2
**Confidence:** 4

**Summary:**

This paper argues that ERM often fails to generalize because the optimization process is drawn to the strongest features in the training distribution — and those strongest features are frequently spurious. To address this, they propose to (i) estimate which training examples are supporting overly strong (possibly spurious) features using a datamodel-style influence approximation, and then (ii) re-train with a sample-weighted ERM that down-weights examples tied to the strongest features. This procedure, termed Feature Strength Regularization (FSR), improves worst-group / OOD accuracy on several standard robust/OOD benchmarks, often without requiring group labels.

**Strengths:**

- The paper pinpoints the optimization bias toward strong-but-spurious features, which is intuitive and empirically plausible (supported by synthetic experiments)
- The final training objective is a weighted ERM, which is easy to implement
- The proposed FSR method does not rely on group labels.

**Weaknesses:**

- This paper focuses on the cases where the dataset contains many strong-but-spurious features. What if the invariant features are just slightly stronger than the invariant features? Will FSR hurt the OOD performance in this case since it would down-weights the invariant features?
- The idea of computing feature strength based on marginal influence $I_c(k)$ and the datamodel approximation is already proposed in Khaddaj et al. (2023). This reduces the novelty of FSR.
- To my understanding, the sample reweighing scheme in FSR can only tackle certain type of covariate shifts. The capability boundary of FSR is not clearly delineated. The authors may consider the simple two-bit-env setup used in "Does Invariant Risk Minimization Capture Invariance?" and "Pareto Invariant Risk Minimization: Towards Mitigating the Optimization Dilemma in Out-of-Distribution Generalization", which studies the capability boundary of IRMv1.
- The experimental improvement is marginal in several benchmarks

Minors:
- Equation (2) contains typos

**Questions:**

See weaknesses

---

### Note · Authors · 2025-11-30

I have read and agree with the venue's withdrawal policy on behalf of myself and my co-authors.